PLOS · Biology

# Structure of human glycoprotein 2 reveals mechanisms underlying filament formation and adaption to proteolytic environment in the digestive tract

Jianting Han[1], Meinai Song[1], Yijia Cheng[1], Wei Gong[2], Fei Zhang[2]*, Qin Cao[1]*

**1** Bio-X Institutes, Key Laboratory for the Genetics of Developmental and Neuropsychiatric Disorders, Ministry of Education, Shanghai Jiao Tong University, Shanghai, China, **2** Department of General Surgery, Xinhua Hospital Affiliated to Shanghai Jiao Tong University School of Medicine, Shanghai Research Center of Biliary Tract Disease, Shanghai, China

* caoqin@sjtu.edu.cn (QC); zhangfei@xinhuamed.com.cn (FZ)

## Abstract

Glycoprotein 2 (GP2) and Uromodulin (UMOD) are considered as paralogs that share high sequence similarity and have similar antibacterial functions. UMOD are abundant as filaments in the urinary tract, and a high-mannose N-glycosylation site located on the N-terminal region protruding from UMOD filament core (referred to as branch) acts as an adhesion antagonist against pathogenic bacterial infections. The antibacterial function of UMOD can be eliminated by proteases, as the UMOD branch is susceptible to proteolytic activity. GP2 is expressed in the pancreas and secreted into the digestive tract. Whether GP2 executes its function in filament form and how it remains functional in the protease-enriched digestive tract is unclear. In this study, we extract GP2 filaments from surgically excised human pancreas and determined their cryo-EM structure. Our structure analysis unveiled that GP2 forms filaments with its ZP modules, composing the ZPN and ZPC domains along with a linker that connects these two domains. The N-terminal region (branch) of GP2 does not constitute the filament core and appears flexible in the cryo-EM structure. Our biochemical experiments suggested that although the GP2 branch is also protease-susceptible, additional high-mannose N-glycans were identified on the protease-resistant GP2 filament core. Consequently, the branch-free GP2 filaments retain their binding ability to the bacterial adhesin FimH, ensuring GP2's antibacterial function unaffected in the proteolytic environment. Our study provides the first experimental evidence of GP2 filament formation and reveals the molecular mechanisms underlying GP2's adaptation to a different environment compared to UMOD.

**Data availability statement:** Cryo-EM map and atomic model of human GP2 filaments present in this study have been deposited into the Worldwide Protein Data Bank (wwPDB) and the Electron Microscopy Data Band (EMDB) with accession codes PDB 8XC5 and EMD-38237. Raw data files of LC–MS/MS analysis were deposited in ProteomeXchange database under accession of PXD055045.

**Funding:** This work was supported by the Ministry of Science and Technology of China (STI2030 major projects 2022ZD0212500 to Q.C) and National Natural Science Foundation (NSF) of China (32271276 to Q.C), China Postdoctoral Science Foundation (No. 2023M732334 to F.Z.) and Shanghai Jiao tong University Medical-engineering cross Program (No. YG2024QNB14 to F.Z.). The funders had no role in study design, data collection and analysis, decision to publish, or preparation of the manuscript.

**Competing interests:** The authors have declared that no competing interests exist.

**Abbreviations:** FDR, false discovery rate; FSC, Fourier shell correlation; GP2, glycoprotein 2; UMOD, uromodulin; UPEC, uropathogenic *Escherichia coli*; ZP, zona pellucida.

## Introduction

Proteins execute their physiological functions in various forms, including extended fibrillar polymers such as cell skeletons [1], cilia [2], and cytoophidia [3]. A ubiquitous class of secret proteins can form extracellular filaments using their bipartite zona pellucida (ZP) modules, playing crucial roles in fundamental biological processes, including hearing, fertilization, and antibacterial defense [4]. The glycoprotein uromodulin (UMOD) is the most extensively studied member in this protein class. It is the most abundant protein in urine and protects the urinary tract against bacterial infections [5,6]. To adhere to uroepithelial cells, uropathogenic *Escherichia coli* (UPEC) uses the adhesin FimH located at the tip of its type 1 pili to recognize high-mannose N-glycans present on the uroepithelial receptor uroplakin 1a [7,8]. UMOD acts as an adhesion antagonist for UPEC through its high-mannose N-glycosylation at Asn275 [8,9]. Cryo-EM studies of UMOD filaments derived from human urine have revealed that UMOD forms zigzag-shaped filaments with intertwined ZP domains (ZPN, ZPC) and an extended linker in between [10,11]. The UPEC adhesion antagonist site, N-glycosylated Asn275, is located on the flexible branch protruding from the filament core [10,11]. This architecture enables UMOD filaments to encapsulate UPEC through multivalent interactions, facilitating pathogen aggregation and clearance [8].

Glycoprotein 2 (GP2) shares a high degree of sequence similarity with UMOD (53% sequence identity), and has been recognized as a paralog of UMOD resulting from gene duplication [12]. GP2 is expressed and secreted by the pancreas and serves a similar antibacterial function in the digestive tract as UMOD does in the urinary tract [9,12–14]. The human GP2 is predicted to contain 10 N-glycosylation sites, and Asn65 on its branch has been identified as a high-mannose site, presumed to act as an adhesion antagonist, and the molecular insights of the Asn-65 containing branch of GP2 has been revealed [9]. Given that GP2 possesses a complete ZP module, it is believed that GP2 can also form filaments in a manner similar to UMOD. However, to date, there is a lack of experimental evidence regarding GP2 filament formation. Moreover, previous studies have indicated that the branch of UMOD can be cleaved from its filament core by elastase, resulting in branch-free UMOD filaments (eUMOD) that do not bind to the UPEC adhesin FimH. Whether GP2's branch is also susceptible to proteases and how GP2 maintain its antibacterial function in the protease-rich environment of the digestive tract remains unclear. In this study, we extracted GP2 filaments from surgically excised human pancreas tissue and determined their cryo-EM structure at 3.5 Å resolution. Our structural and biochemical experiments suggest that, similar to UMOD, GP2 forms filaments with its ZP modules, and its branch is susceptible to elastase. Meanwhile, GP2 develops additional high-mannose N-glycosylation sites on its filament core to uphold its antibacterial function in the face of proteolytic pressures within the digestive tract.

## Results

### GP2 filaments extraction and cryo-EM structure determination

To obtain human-derived GP2 filaments, we collected surgically excised pancreas tissue from a patient with pancreatic masses (donor 1), which were later diagnosed

as a benign tumor. Filaments were extracted from the tumor-free pancreas tissues and examined using electron microscopy (EM, Fig 1a). Cryo-EM data were collected, and analysis of the 2D classes revealed a zigzag-shaped filament species similar to UMOD filaments [10,11] (Figs 1b and S1). In addition, the cryo-EM dataset contained another filament species that may represent B-DNA, similarly as previously reported [15], as well as globular particles that likely represent proteins co-extracted with the filaments and may have been damaged during extraction or data collection (Fig 1a and 1b). A cryo-EM map at a resolution of 3.5 Å was generated for the zigzag-shaped filaments through single-particle 3D-reconstruction (S2a–S2d Fig), and the atomic model can be successfully built with the human GP2 sequence confirmed by genotyping (see the next paragraph), with most side chain densities well explained (S2f Fig). The cryo-EM data collection and processing statistics are summarized in Table 1.

To date, four human GP2 isoforms have been identified [16,17], likely resulting from alternative splicing (Fig 2a). To investigate the specific GP2 isoform present in this donor, genotyping was performed, revealing the presence of isoforms 3 and 4 in the sample derived from donor 1 (Fig 2a, S1 Table). Given that the sequence within the filament core remains

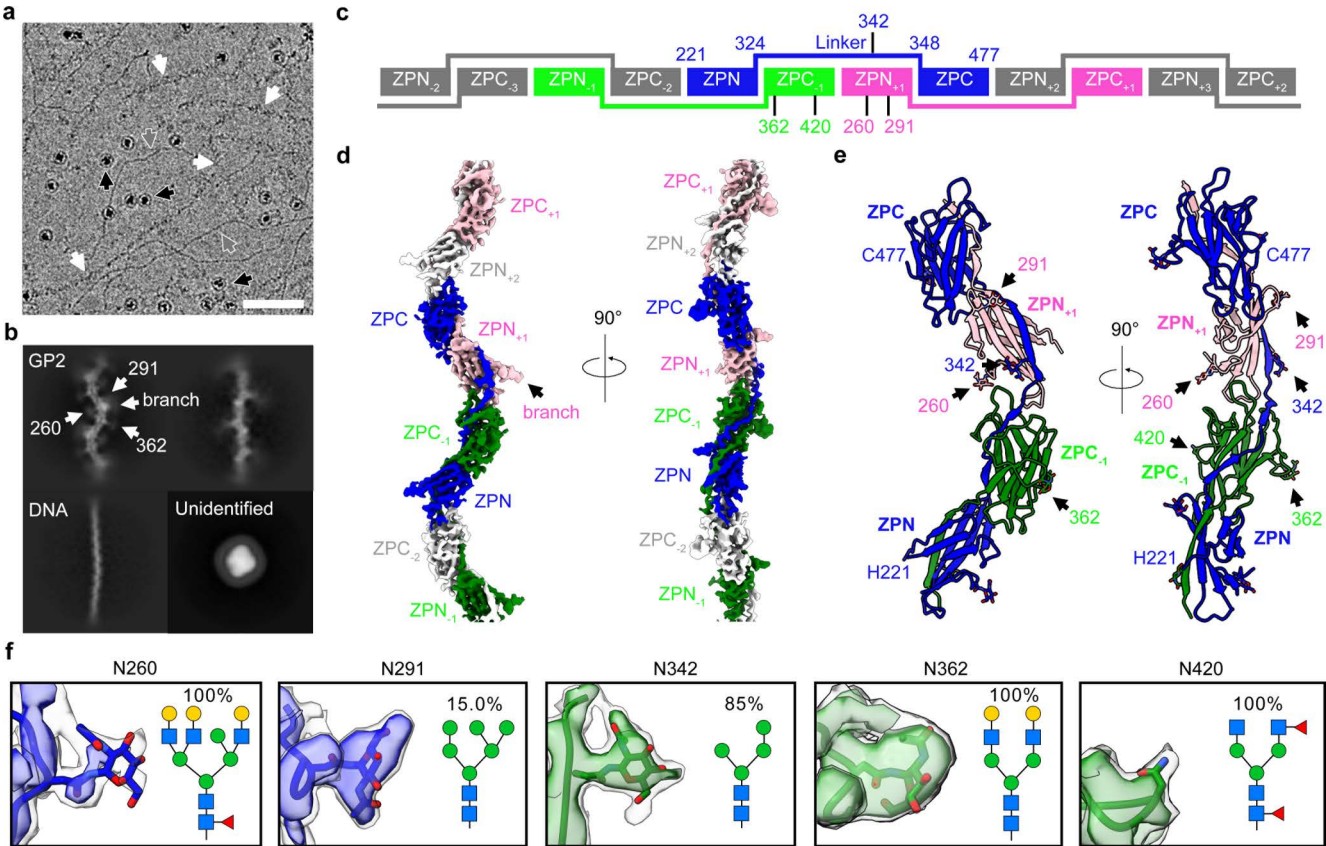

**Fig 1. Cryo-EM structure of GP2 filaments. a,** A representative cryo-EM micrograph of filaments extracted from the human pancreas. GP2 filaments are indicated by white arrows, DNA filaments by grey arrows, and unidentified globular proteins by black arrows. Scale bar represents 50 nm. **b**, Representative 2D classes of GP2, DNA, and unidentified globular proteins. White arrows point to the N-glycosylation sites and the branch (contains the D10C domain and EGF-like domain shown in Fig 2a). **c**, Schematic of GP2 filaments, with labeled residue ranges of ZP domains and four N-glycosylation sites. **d**, Cryo-EM map of GP2 filaments, color-coded according to the schematic in panel c. The density of one of the flexible branches was indicated by a black arrow. **e**, Atomic model of GP2 filaments, with five N-glycosylation sites marked by arrows. The orientation of GP2 filaments in panels **b, d, e** is the same. **f**, Maps, models, and predominant glycan types (highest abundance, except for Asn291, which ranks as the second highest) at the five glycosylation sites within the filament core as identified through mass spectrometry. The maps in higher threshold are colored in blue or green, while those with lower threshold are colored in grey. The data underlying this figure can be found in S7–S15 Figs and S3 Table.

**Table 1. Cryo-EM data collection, refinement and validation statistics of GP2 filaments.**

| | GP2 (EMD-38237, PDB 8XC5) |
|---|---|
| **Data collection and processing** | |
| Magnification | ×130,000 |
| Voltage (kV) | 300 |
| Electron exposure (e⁻/Å²) | 40 |
| Defocus range (μm) | 0.8–3.5 |
| Pixel size (Å) | 1.05 |
| Symmetry imposed | $C_1$ |
| Initial particle images (no.) | 1,437,210 |
| Final particle images (no.) | 249,718 |
| Map resolution (Å) | 3.5 |
| FSC threshold | 0.143 |
| Map resolution range (Å) | 200–3.5 |
| **Refinement** | |
| Initial model used (PDB code) | *De novo* |
| Model resolution (Å) | 4.1 |
| FSC threshold | 0.5 |
| Model resolution range (Å) | 200–4.1 |
| Map sharpening *B* factor (Å²) | 190 |
| Model composition | |
| Nonhydrogen atoms | 1,892 |
| Protein residues | 233 |
| Ligands | 4 |
| *B* factors (Å²) | |
| Protein | 64.4 |
| Ligand | 46.8 |
| R.m.s. deviations | |
| Bond lengths (Å) | 0.006 |
| Bond angles (°) | 1.167 |
| **Validation** | |
| MolProbity score | 2.63 |
| Clashscore | 31.3 |
| Poor rotamers (%) | 0 |
| Ramachandran plot | |
| Favored (%) | 85.7 |
| Allowed (%) | 14.3 |
| Disallowed (%) | 0 |

consistent across all four isoforms, we maintain that the presence of two isoforms in donor 1 does not impact the analysis of the filament structure of GP2. Meanwhile, no mutations have been detected in the GP2 sequence of donor 1. It is important to acknowledge that while isoforms 3 and 4 contain missing residues, we will adhere to the numbering of GP2 isoform 1 for the subsequent analysis to maintain consistency.

### Cryo-EM structure of GP2 filament core

In the cryo-EM structure of GP2, the bipartite zona pellucida (ZP) modules (residues 221−478) comprise the filament core (Fig 2a). The D10C and EGF-like domains extend outward from the filament core as flexible branches, and no

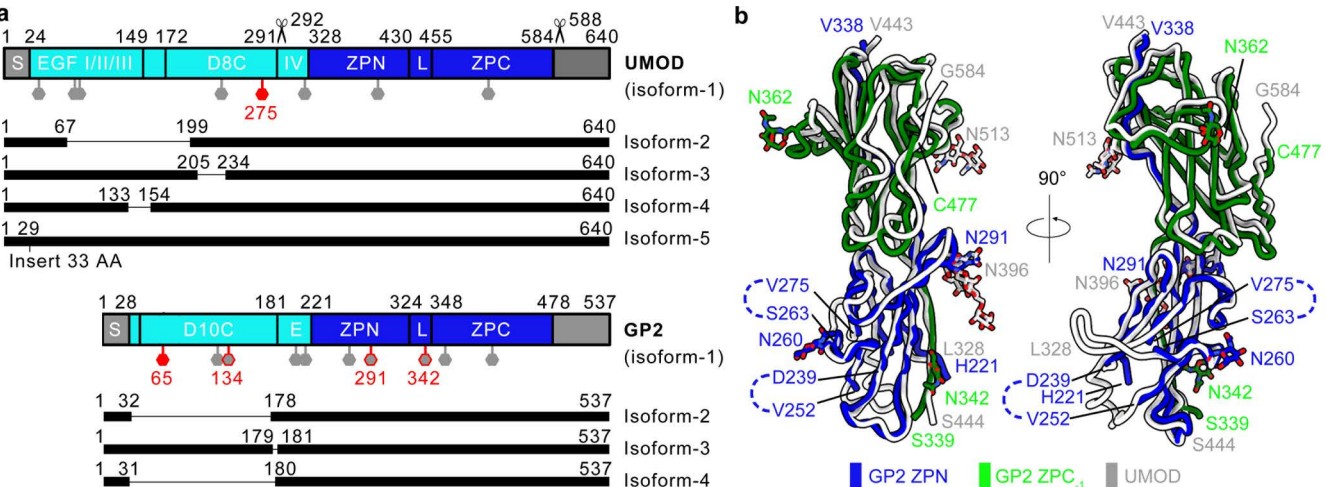

**Fig 2. Structural and sequential comparison between UMOD and GP2. a,** Domain structures of UMOD and GP2, including their reported isoforms (see "Discussion" for details)**.** The filament cores are colored in blue, the branches are colored in cyan, and the other regions are colored in grey. S, signal peptide; IV, EGF IV domain; L, linker; E, EGF-like domain. The elastase and hepsin cleavage sites reported in UMOD were indicated as scissor labels. N-glycosylation sites are represented as hexagons. Previously identified as high-mannose glycans are shown in red, while high-mannose glycans identified in this study (contain at least 15% high-mannose glycans if multiple glycans were detected) are colored in grey with a red outline. **b,** Structure superimposition of GP2 and UMOD (PDB ID 6ZS5). The data underlying this figure can be found in S7–S15 Figs and S3 Table.

high-resolution densities are observed around these regions (Figs 1b, 1d, 2a, and S2g). These findings are consistent with previously reported UMOD structures [10,11] (Fig 2a). Additionally, two loops on the ZPN domains (residues 239−251 and 264−274) are also flexible and lack well-defined densities, likely because their conformations are influenced by the flexible branch regions (S2e and S2g Fig). The filament core structure of GP2 closely resembles that of UMOD [10,11] (Fig 2b), including the secondary structures within each domain (S3 Fig), as well as the overall arrangements of domains (Fig 1c–1e). Each ZP module within GP2 filaments is symmetrically related to the adjacent module with a helical twist of approximately 177.2 degrees and a helical rise of approximately 66.3 Å. To form filaments, the ZPN and ZPC domains from a given ZP module are separated by an extended linker (residues 324−348), a ZPC domain from the previous ZP module (ZPC$_{-1}$), and a ZPN domain from the subsequent ZP module (ZPN$_{+1}$, Fig 1c–1e). These regions assemble through a β-sheet complementation mechanism, where the first and second β-strands in linker region (L1 and L2) complemented the fold of ZPC$_{-1}$, and the third β-strand in linker region complemented the fold of ZPN$_{+1}$ (S3b and S4a Figs). The side chain interactions between linker region, ZPC$_{-1}$, and ZPN$_{+1}$ are predominantly hydrophobic, with some additional electrostatic interactions (S4 Fig). These interactions, alone with five pairs of intra-domain disulfide bonds, contribute to the stability of GP2 filaments, similar to UMOD [8,10,11].

The mature human UMOD has been documented to terminate at Phe587, a consequence attribute to hepsin cleavage at the Arg588–Ser589 site followed by subsequent C-terminal trimming (Figs 2a and S3a) [18]. Structural comparisons between mature UMOD filament and immature pro-UMOD suggest that this cleavage is essential for filament assembly because the uncleaved C-terminal pro-peptide occupies the binding pocket of the linker region and thus disturbs filament formation [10]. In the case of GP2, a similar mechanism of C-terminal cleavage is anticipated based on the following observations: (i) sequence alignment reveals the presence of a comparable hepsin cleavage site within GP2, akin to that found in UMOD (S3a Fig); (ii) structural superimposition of the cryo-EM structure of GP2 filament and the AlphaFold model of full-length GP2 demonstrates that the C-terminal pro-peptide aligns with the position of the linker region (S3d Fig), similar to UMOD. However, it is challenging to experimentally confirm the cleavage of the C-terminal pro-peptide of GP2 in this study, primarily due to the limited purity of GP2 isolated from human tissues.

We observed extra densities for four N-glycosylation sites on the filament core of GP2 (Asn260, Asn291, Asn342, and Asn362, Fig 1f). The densities of three of these N-glycosylation sites were observed already in 2D classes (Fig 1b, Asn260, Asn291, and Asn362, see white arrows). However, in the cryo-EM map, we only observed densities corresponding to a monosaccharide at each site (Fig 1f). This is probably due to the intrinsic flexibility of these N-glycans, which may facilitate pathogen encapsulation of GP2 filaments. Mass spectrometry analysis confirmed the presence of N-glycosylation at Asn420 (described later), although it was not observed in the cryo-EM map (Fig 1f). By integrating the insights from the cryo-EM map and mass spectrometry data, we built a monosaccharide molecule (*N*-acetyl-β-d-glucosamine, NAG), the sole monosaccharide available for the initial position of asparagine linked glycosylation [19,20] on each of the four sites mentioned above (Fig 1f). Interestingly, despite the high sequence and structure similarity between GP2 and UMOD filaments, the observed N-glycosylation sites are not entirely identical. While GP2 has five glycosylation sites on its filament core, UMOD has two, with only one site overlapping between the two proteins (Asn291 on GP2 versus Asn396 on UMOD, Figs 2a, 2b and S3a). These findings suggest that the glycosylation sites on GP2 and UMOD filament cores may have been subjected to different selective pressure during evolution.

## Proteolytic resistance of GP2 filaments

Previous studies have suggested that the filament core of UMOD is resistant to proteases like elastase due to its tightly packed structure, whereas its branch can be cleaved at position Ser292, resulting in filaments without branches, referred to as eUMOD [8,21]. To investigate whether GP2 filaments exhibit similar proteolytic property to UMOD, we incubated GP2 filaments extracted from donor 1 with elastase, and probed the samples using GP2 antibodies targeting either the branch (residues 35–179, referred to as the branch antibody in the following discussion) or regions covering part of the branch and the filament core (residues 111–387, referred to as the filament core antibody in the following discussion; Figs 3a and S5).

In the sample prior to elastase cleavage, we observed a 105 kDa band recognized by both antibodies (Fig 3a). Mass spectrometry analysis confirmed that this band corresponds to GP2 containing residues 28–524, representing the near full-length form of GP2 isoform 3 (S2 Table, also see "Methods" and the next subsection, detail analysis in S1 Supplementary Notes (Note 1)). Additionally, we detected a 70 kDa band in the same sample, visible only with the filament core antibody, and bands ranging from 15–25 kDa that were only visible with the branch antibody (Fig 3a). These findings suggest that the 70 kDa band likely corresponds to the branch-cleaved fractions of GP2 (or the branch-free isoform 4 of GP2), while the 15–25 kDa bands may represent free and partially shredded branches (S2 Table, S5a Fig, detailed analysis in S1 Supplementary Notes (Note 1)). Due to the presence of the 15–25 kDa bands, we believe that the 70 kDa band is not solely composed of GP2 isoform 4 but is at least partially derived from branch cleavage. In other words, these observations indicate that the branch cleavage may have occurred in a subset of GP2 filaments even before elastase treatment. However, it remains uncertain whether this cleavage occurred in the pancreas or during filament extraction. It is also important to note that at the current stage, we are unable to pinpoint the exact cleavage sites on GP2 responsible for branch cleavage.

Following the incubation with elastase (referred to as eGP2 samples in the following discussion), we observed the disappearance of the 105 kDa band, indicating the complete cleavage of GP2 filaments (Fig 3a). No bands positive for the branch antibody remained, except for a faint 10 kDa band at the migration front of the SDS–PAGE, suggesting all branches are cleaved and completely shredded into pieces, similar to the behavior observed with UMOD [8] (Fig 3a). The 70 kDa band was also absent in this sample, and instead, smaller bands were only visible with the filament core antibody (Fig 3a), indicating the presence of additional cleavage sites on the GP2 filament core (S2 Table, S5b Fig, detailed analysis in S1 Supplementary Notes (Note 2)). Despite these additional cleavage sites, our cleavage assay demonstrated that the filament core of GP2 is resistance to elastase. The reason is that, although there are numerous elastase cleavage sites in the GP2 sequence, bands above the migration front of SDS–PAGE (such as those around 35 kDa) were

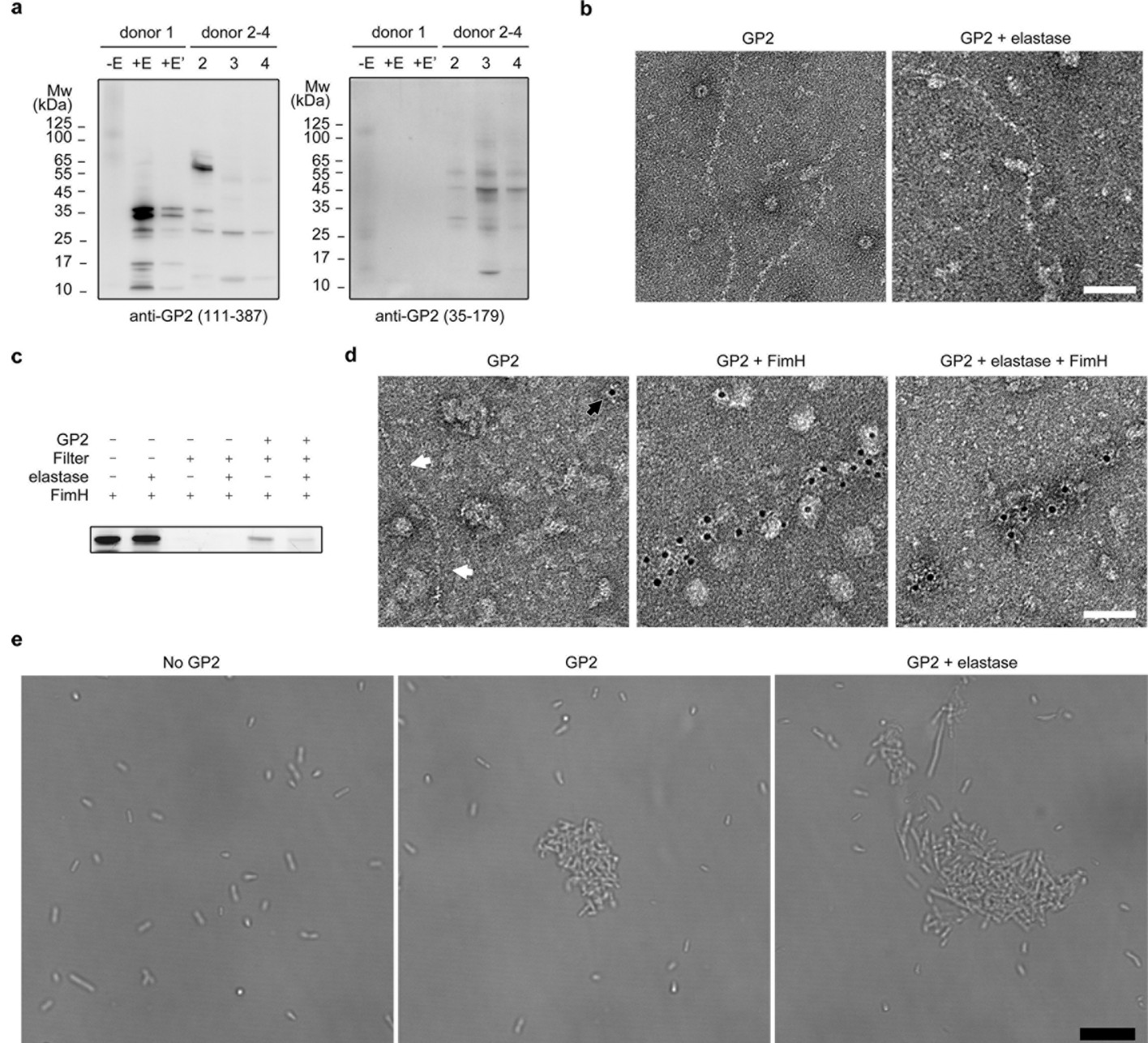

**Fig 3. Biochemical characterization of GP2 filaments. a**, Western blotting analysis of GP2 derived from human pancreas (donor 1, before and after elastase cleavage) and small intestines (donors 2–4, without elastase treatment), probed with two GP2 antibodies, anti-GP2 (35–179) and anti-GP2 (111–387). −E, GP2 incubated without elastase; +E, GP2 incubated with elastase; +E', the same sample as +E, but the loading volume for SDS–PAGE was reduced to one-fifth; Mw, molecular weight. **b**, Representative negative-stain EM images of (e) GP2 derived from the pancreas. Scale bar represents 50 nm. **c**, The SDS–PAGE of interaction assays between FITC-FimH$_L$ and GP2. The bands represent FITC-FimH$_L$ that were visualized with fluorescence. Samples were either washed using centrifugal filters (+ in the row labeled "filter") or directly analyzed using SDS–PAGE without wash (− in the same row) as controls. **d**, Representative negative-stain EM images of (e) GP2 incubated with or without FimH$_L$, labeled with immunogold and FimH antibodies. White arrows label GP2 filaments and the black arrow labels immunogold beads. Scale bar represents 50 nm. **e**, Representative light microscopy images of (e) GP2-mediated *E. coli* cell aggregation. Scale bar represents 10 µm. All assays were repeated independently at least three times with similar results. The data underlying this figure can be found in S1 Raw Images.

observed, suggesting that the quaternary structure of GP2 filament core remains intact, protecting these cleavage sites. To further validate the proteolytic resistance of GP2 filament core, we examined the (e)GP2 samples using negative stain EM and found that filaments with similar morphology were present in both samples (Fig 3b).

Taken together, our proteolytic cleavage assays demonstrated that, like UMOD, the branch of GP2 can be cleaved and shredded into small pieces, whereas the filament core of GP2 is resistant to elastase cleavage, potentially resulting in branch-free GP2 filaments. We note that the GP2 filament samples utilized in this study contains some contaminated proteins, and it is challenging to precisely determine the purity of GP2 in these samples (see "Methods"). Hence, while less probable, we cannot entirely rule out the potential impact of the contaminated proteins on the outcomes of our cleavage assay.

## Identification of N-glycosylation types of GP2 filaments

Previous studies have reported that the branch-free eUMOD filaments do not bind to FimH or piliated UPEC, because the high-mannose N-glycosylation site, Asn275, was located on the branch [8] (Fig 4). The branch of GP2 is also susceptible to proteolytic cleavage, we therefore reasoned that GP2 filament core should carry additional high-mannose N-glycans to maintain its antibacterial functions in protease-rich digestive tract. Considering this, we carried out N-glycosylation analysis using liquid chromatography-tandem mass spectrometry (LC–MS/MS). The mass spectrometry data was analyzed using a GP2-only database (containing the sequences of GP2 isoform 3 and isoform 4), leading to the identification of nine N-glycosylation sites from pancreas derived GP2 filaments (Asn122, Asn134, Asn204, Asn216, Asn260, Asn291, Asn342, Asn362, and Asn420, also see S3 Table and S7–S15 Figs). Although Asn65 had been previously identified to be a high-mannose N-glycosylation site of GP2 [9], it was not detected in this study. We note that the absence of Asn65 detection in our sample does not necessarily indicate the lack of N-glycosylation at that site, as the glycopeptides we

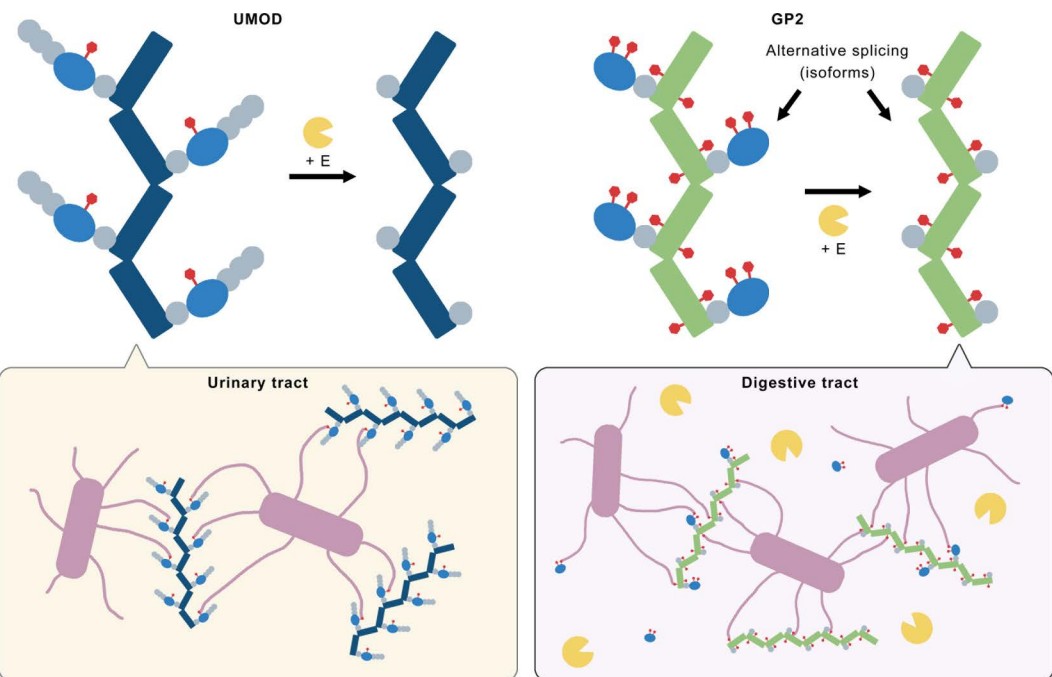

**Fig 4. Proposed model of GP2 filaments' function in the digestive tract.** The filament core of UMOD and GP2 is colored in dark blue and green, respectively. The EGF domains and D8C/D10C domains in the branches of both proteins are shown in grey and light blue, respectively. High-mannose N-glycans are colored in red. UPEC cells and their adhesive pilus are colored in pink, while the proteases (such as elastase) are colored in yellow.

detected in mass spectrometry did not cover the region around Asn65 (see "Methods"). Additionally, N-glycosylation at Asn420 was detected through LC–MS/MS analysis, yet its cryo-EM density was lacking, even for the first monosaccharide unit (Fig 1f). A potential explanation for this absence is that the population of glycosylation at this site is limited. Among the identified N-glycosylation sites, three contain high-mannose type glycans (Asn134, Asn291, and Asn362), with Asn291 and Asn362 located on the filament core (Figs 1f and 2a). In Asn291, two high-mannose types rank among the top 5 abundant glycan types, collectively constituting 21.6% of the total glycan population; in Asn342, both detected glycan types are high-mannose (Fig 1f, S3 Table). These findings suggested that, unlike UMOD, the branch-free GP2 filament core should be capable of binding to FimH and functioning as an antibacterial antagonist.

Considering that the sample extracted from the pancreas tissue was not solely GP2, the presence of contaminated glycoproteins could potentially influence the outcomes of N-glycosylation identification. To address this concern, we analyzed the mass spectrometry data using human proteome database (with GP2 sequence replaced by those of GP2 isoform 3 and isoform 4), a commonly employed target-decoy setting for heterogeneous samples. With this configuration, we can still identify several previously mentioned N-glycans, including two high-mannose types on Asn291, exhibiting a comparable relative percentage of the total glycan population (S3 Table). Additionally, certain N-glycans could only be identified using the GP2 sequences and not the human proteome database, likely due to the restricted abundance of these glycopeptides, stemming from the impurity of the GP2 samples. To validate the reliability of these N-glycans further, we analyzed the mass spectrometry data using GP2-free database (where the GP2 sequence was manually omitted from the human proteome database) and compared the results to those obtained using the GP2-only database. A total of 415 glycopeptides were identified using the GP2-free database. When compared them to the 93 glycopeptides solely detected using the GP2-only database, we identified only 2 pairs of glycopeptides that were detected through either database and shared similar mass (with a mass error less than 20 ppm). Upon manual examination of both pairs, we observed that glycopeptides within each pair exhibited entirely distinct MS/MS spectra (S7 and S14 Figs). These results exclude the effect of contaminated proteins on glycopeptides identification and affirm the reliability of these detected N-glycans.

We observed that the mass spectrometry analysis detected the presence of the C-terminal transmembrane domain region of GP2 (see "Methods"). This suggests that the sample contains a certain amount of nascent GP2 that has not yet exited the endoplasmic reticulum (ER, detailed discussions see S1 Supplementary Notes (Note 3)). We note that due to the GP2 sample being extracted from the human tissue, complete elimination of ER protein contamination is challenging. However, the identification of almost all N-glycosylation sites predominately with Golgi-processed glycan types indicates that the ER form of GP2 was not the main species in our sample (detailed analysis see S1 Supplementary Notes (Note 3)). Therefore, the N-glycosylation types identified in this study are likely representative of the mature form of GP2.

### Interaction of GP2 filaments with *E. coli* adhesin FimH

We examined the binding ability of (e)GP2 filaments to the lectin domain of FimH (FimH$_L$). We incubated fluorescence-conjugated FimH$_L$ (FITC-FimH$_L$) with (e)GP2 filaments derived from donor 1 and removed the unbound FITC-FimH$_L$ using a 100 kDa cut-off centrifugal filter. The results demonstrated that FITC-FimH$_L$ was retained during centrifugation when incubated with both GP2 and eGP2 (Fig 3c). After elastase cleavage, the amount of FITC-FimH$_L$ retained during centrifugation decreased compared to that without cleavage (Fig 3c). We believe it is reasonable based on our previous analysis of N-glycosylation types. There are four high-mannose glycosylation sites (Asn65, Asn134, Asn291 and Asn342) on GP2. The elastase cleavage removes two high-mannose sites (Asn65 and Asn134) located on the branch from the filament, resulting in a weaker binding between eGP2 filaments and FITC-FimH$_L$. Additionally, the cleaved branch is expected to be fully shredded and release free peptides containing high-mannose glycosylated Asn65 or Asn134, which may compete with the binding of FITC-FimH$_L$ to eGP2 filaments, further contributing to the apparent weaker binding of eGP2. In summary, our interaction assays shown in Fig 3c suggest that FimH$_L$ binds to both GP2 and eGP2.

To visualize the interaction between FimH$_L$ and (e)GP2, we stained the FimH$_L$-(e)GP2 mixture using immunogold and FimH antibody. We found that both GP2 and eGP2 filaments were labeled with immunogold when incubated with FimH$_L$, whereas GP2 filaments without FimH$_L$ were not labelled (Fig 3d). These findings suggested a direct binding between FimH$_L$ and GP2 filaments, and the branch-free GP2 filament core retained the ability to bind FimH$_L$. We then investigated whether (e)GP2 filaments could induce the aggregation of *E. coli* cells, similar to UMOD [8]. When GP2 or eGP2 filaments were incubated with *E. coli* cells, the formation of bacterial clumps was observed, in contrast to (e)GP2-free *E. coli* cells (Fig 3e). The GP2-induced bacterial clumps can be eliminated through competition with D-mannose in a dose-dependent manner (S6a Fig), suggesting that these clumps stem from FimH binding. Taken together, these results indicated that both GP2 and eGP2 filaments possess functional properties in facilitating pathogen aggregation.

**GP2 derived from human small intestine**

To investigate the cleavage of GP2 in the human digestive tract, we conducted western blotting using wash-out solutions from small intestine tissues obtained from three donors. The results revealed the absence of the 105 and 70 kDa bands in these samples, indicating that GP2 undergoes proteolytic cleavage upon secretion from the pancreas into the small intestine (Fig 3a). The presence of bands positive for the filament core antibody above the migration front of SDS–PAGE (around 10 kDa) suggests that the filament core remains intact in the small intestine. However, the cleavage pattern of GP2 derived from the small intestine differed from that derived from the pancreas. In all three small intestine samples, we observed bands that were positive for the branch antibody and were above 35 kDa, indicating the presence of intact branches, possibly still connected to the filament core (S2 Table, S5c and S5d Fig, detailed analysis in S1 Supplementary Notes (Note 4)). These findings suggest that the cleavage of branches in these samples were not as extensive as observed in our elastase cleavage assays. This difference may be attributed to the dynamic balance between protease activity and GP2 secretion in the small intestine (see "Discussion"). Notably, in donor 2, we observed a predominant band around 62 kDa that was only visible with the filament core antibody. This observation suggests that this band likely represents the branch-free filament core (detailed analysis in S1 Supplementary Notes (Note 4)). Furthermore, it indicates that the branch-free GP2 filaments constitute the main population in the small intestine of donor 2, underscoring the importance of high-mannose N-glycosylation sites on the filament core for protection against bacterial infections in this particular case.

## Discussion

Proteins containing bipartite zona pellucida modules have been found to play crucial roles in various biological processes, often functioning in the form of filaments [4]. Despite this knowledge, the molecular mechanism underlying the assembly of ZP module filaments remained unclear until recent studies on UMOD filaments [10,11], as well as the crystal structure of pseudo-infinite filaments of digested fish vitelline envelope proteins [22]. Whereas it remains to be determined whether other ZP module proteins assemble into filaments using a similar mechanism. In this study, we focused on GP2, a ZP module protein considered to be the paralog of UMOD, and derived from the human pancreas. Using cryo-EM, we determined the structure of GP2 filaments. Our cryo-EM analysis revealed that GP2 forms filaments that similar to UMOD, featuring a tightly packed filament core and flexible branches. These findings raise intriguing questions regarding the functional mechanism of GP2 filaments in the digestive tract, which is known for its abundance of proteolytic activities.

Generally, the purpose of generating two paralog proteins through gene duplication is to allow them to evolve independently and acquire distinct functions or adopt to different environments. Previous studies have suggested that GP2 and UMOD execute similar antibacterial functions in the digestive tract and the urinary tract, respectively [13,14]. Therefore, it can be inferred that the primary reason of distinguishing GP2 and UMOD is to enable these proteins to function in different environments. It have been demonstrated that elastase can cleave the branches containing the adhesion antagonist site from the UMOD filament core, resulting in the formation of eUMOD filaments that do not bind to FimH or piliated

UPEC [8]. As the digestive tract is rich in active proteases, including elastase, GP2 has to evolve differently to maintain its FimH-binding ability in a protease-rich environment.

Sequence alignment analysis revealed that the cleavage site (Ser292) used for UMOD branch cleavage remained unchanged in GP2 (S3a Fig). Our cryo-EM structure demonstrated that the branch region of GP2, similarly to UMOD, is flexible without high-resolution cryo-EM density, indicating that this cleavage site in GP2 is not protected by ordered protein folding (Figs 1b, 1d, and 1e, S2e and S2g, S5e and S5f). These findings suggest that the branch of GP2, like UMOD, can be cleaved from the filament core. Our elastase cleavage assays supported this hypothesis, showing that pancreas-derived GP2 filaments can be cleaved by elastase, resulting in smaller peptides that are branch-free while maintaining the filament structure (Fig 3a and 3b). Considering the vulnerability of the branch in a protease-rich environment, GP2 may need to carry out its antibacterial functions independently of the branches. To further investigate this, we analyzed the N-glycosylation pattern of pancreas-derived GP2. Interestingly, unlike UMOD, we discovered two high-mannose N-glycosylation sites, Asn291 and Asn342, on the GP2 filament core (Figs 1f, 2a, S7–S15, S3 Table), which are known to act as receptors for the UPEC adhesin FimH [23]. Our interaction assays confirmed that both branch-connected and branch-free GP2 filaments are capable of binding to FimH$_L$ and inducing the aggregation of *E. coli* cells (Fig 3c–3e). These observations indicate that the key difference between GP2 and UMOD lies in the ability of their branch-free filament cores to bind FimH. We believe this difference primarily stems from the distinct functional environment of GP2 and UMOD. The protease-rich environment likely acted as a selective pressure during evolution, leading to the development of high-mannose N-glycans in the GP2 filament core.

It is important to acknowledge that the N-glycans identified in this study may not represent the complete types present on GP2. Some N-glycans could have undergone degradation during filament extractions, mass spectrometry sample preparations, or could fall below the detection sensitivity of the mass spectrometry analysis due to their limited abundance. Hence, the reported results of N-glycan identification may exhibit survivor bias. However, we maintain that our conclusion drawn from the N-glycan identification remains unchanged. Although these high-mannose types identified on the GP2 filament core may not encompass the complete spectrum of N-glycans at each site, they likely represent the most stable and prevalent ones. Consequently, GP2 can utilize these high-mannose types to bind to FimH and fulfill its function without the branch. It is also worth noting that the quality of our MS/MS data is not optimal, as the spectra indicate insufficient fragmentation for the majority of glycopeptides. The quality of our MS/MS data is likely caused by the fact that our sample is the crude extract from human tissue, and that the glycosidic bonds are much weaker than peptide bonds. Therefore, we acknowledge that the identified glycosylation types in this study may not definitive. Nevertheless, we believe this does not impact the conclusion of our study. The overall analysis of the current MS/MS data is reliable, and our biochemical experiments further support the presence of high-mannose glycosylation sites on the filament core of GP2. This is evidenced by the ability of elastase-processed GP2 filaments to bind FimH and induce bacteria aggregation (Fig 3). Meanwhile, we cannot rule out the possibility that N-glycan types on GP2 may differ among individuals or be influenced by an individual's medical background. Further studies are needed to examine the N-glycan types on GP2 extracted from diverse donors with varying medical conditions.

The functionality of the GP2 filaments core is further supported by the presence of branch-free isoforms of GP2. The structural analysis of GP2 filaments suggests that filament architecture primarily depends on the filament core, with the branches extending over the filament backbone. Given that all GP2 isoforms share identical filament core regions, we hypothesize a model where all GP2 isoforms contribute equally to the filament assembly process. Among four reported isoforms of GP2, two are branch-free [16,17] (Fig 2a). In the sample from donor 1 that we examined with genotyping, two GP2 isoforms were identified, one of which is branch-free. These two branch-free isoforms, labeled as isoform 2 and 4, lack residues 32–178 and 31–180, respectively, which encompass most of the branch and all the branch-located N-glycosylation sites (Fig 2a). An recent study has suggested that these two branch-free isoforms are capable of interacting with FimH *in vitro* [24]. These findings support our hypothesis that the GP2 filament core alone is capable of exerting

its antibacterial function, which may explain the survival of these branch-free isoforms throughout evolution. In comparison, for UMOD, no isoforms lacking the region around its high-mannose antagonist site, Asn275, has been reported (Fig 2a).

GP2 is known to be primarily produced in the pancreas and then secreted into the intestine [14]. Our results suggest that GP2 forms filaments before being secreted from the pancreas, and these filaments are stable and resistant to proteases. Taking these findings together, we believe that GP2 in the intestine is likely to maintain its filament structure, except for the potential absence of the branch. In this study, we also investigated the properties of GP2 derived from the small intestines of three donors (Fig 3a). The results confirmed that GP2 undergoes proteolytic cleavage in the digestive tract, as evidenced by the absence of the 105 kDa band (representing uncleaved GP2 isoform 1/3) or the 70 kDa band (representing uncleaved GP2 isoform 2/4). However, in all three samples, the branches did not appear to be completely cleaved (Figs 3a and S5c, S5d). We note that this may represent a dynamic balance between GP2 secretion and protease activities in the digestive tract. The protease activity in the digestive tract is known to oscillate regularly in response to feeding, with fasting leading to a decrease in protease activity. Notably, the small intestine tissues used in this study were obtained from patients undergoing surgical procedures, which required them to fast beforehand. All three small intestine donors had fasted for more than 10 h prior to surgery, suggesting that their GP2, in this case, was under low protease pressure. It is plausible that when the protease activities are high, such as shortly after feeding, GP2 may be more extensively cleaved in the small intestine, similar to what we observed in our *in vitro* cleavage experiment. However, obtaining small intestine tissues shortly after feeding is challenging due to the fasting requirement for surgery, making it difficult to confirm whether the branch of GP2 is fully cleaved when the protease activities are high in the digestive tract. In one of the three small intestine donors (donor 2), we observed a prominent band that appeared to be more likely branch-free (Fig 3a, detailed analysis in S1 Supplementary Notes (Note 4)). Whether this band results from proteolytic cleavage or different isoforms, its presence partially supporting our hypothesis that GP2 may need to function solely with its filament core in the small intestine.

Base the analysis above, we proposed a working model for GP2 function (Fig 4). GP2 and UMOD are paralogs that perform similar antibacterial functions in different environments. UMOD filaments are abundant in the human urinary tract. The functional glycosylation site is exclusively located on the branch of UMOD, so when the branch is cleaved off from the filament core, the remaining eUMOD losses its antibacterial function. Due to limited protease activities in the urinary tract, the physiological function of UMOD is not affected by the vulnerability of the branch (Fig 4, left panel). Similar to UMOD, the branch of GP2 contains the functional glycosylation site and can also be removed through proteolytical cleavage. However, unlike UMOD, GP2 is secreted into the digestive tract, which contains abundant active proteases. In the small intestine, the presence of GP2 branch varies due to the oscillation of protease activities. This means that GP2 needs to execute its function while facing the risk of losing its branch. In addition to proteolytic cleavage, branch-free GP2 can also be produced through alternative splicing [16,17]. Therefore, GP2 has developed additional functional glycosylation sites on its filament core, allowing it to maintain its physiological function even if the branch is completely cleaved (Fig 4, right panel). In other words, we hypothesize that both the branch and filament core of GP2 contain functional glycosylation sites, with those on the filament core acting as backups to compensate for the potential loss of the branch in the protease-rich environment of the digestive tract. It is important to note that when the branches are removed from GP2 filaments, or even further shredded, they retain their high-mannose glycans, enabling them to still bind to FimH. Despite this, it is reasonable to assume that these free branches are less likely to perform antibacterial functions since this function depends on multivalent interactions between GP2 and bacteria [8], where a single GP2 filament can bind to multiple bacteria, and reciprocally, one bacterium can bind to multiple GP2 filaments (Fig 4). Considering that each branch contains only two high-mannose N-glycosylation sites, without even considering the shredding process, their antibacterial efficacy may be compromised.

In summary, in this study, we extracted GP2 filaments from surgical excised human pancreas and determined their cryo-EM structure. Through the analysis of the structure and subsequent biological investigations, we discovered that

despite the sequential and structural similarity between GP2 and UMOD, GP2 employs a previously unreported strategy to adapt to the proteolytic environment of the digestive tract.

## Methods

### Surgical excised human tissues

The pancreas tissue used in this study was obtained from a 32-year-old male patient with pancreatic masses (referred to as donor 1). Pancreatic body and tail resection were performed on this patient, and the subsequent diagnosis suggested that the pancreatic mass was caused by a benign tumor. A tissue sample was collected from the tumor-free area of the excised pancreas. The small intestine tissue samples were obtained from three donors (donors 2−4): donor 2 was a 66-year-old male patient who underwent radical surgery for gastric cancer and colon cancer, donor 3 was a 40-year-old male patient who underwent radical surgery for gastric cancer, and donor 4 was an 83-year-old female patient who underwent pancreaticoduodenectomy for pancreatic cancer. Normal tissues from the tumor-free area of the excised small intestines were used in this study. All tissue samples were stored in −80 °C for future use. The collection of tissues and subsequent procedures were performed with the consent of the patients and in accordance with protocols approved by the Institutional Review Board of Xinhua hospital (approval no. XHEC-D-2025-051). Written informed consent was obtained from the patients at the time of their visit to Xinhua Hospital, agreeing to the application of the samples generated during routine consultations, including outpatient, inpatient and emergency visits, for future studies.

### GP2 extraction

To extract GP2 filaments from the pancreas tissue, the frozen tissues were weighted, diced into small pieces, and resuspended in homogenization buffer (10 mM Tris-HCl, pH 7.4, 800 mM NaCl, 10% sucrose, 0.1% sarkosyl, 1 mM EDTA; 4 ml per gram tissue). The resuspended sample was homogenized on ice using a handheld homogenizer and centrifuged at 21,000 × $g$ at 4 °C for 10 min. The pellet was resuspended in an additional 4 ml per gram tissue homogenization buffer and subjected to another round of homogenization. These homogenization cycles were performed three times, and the homogenized sample was incubated with 2% sarkosyl at 37 °C for 1 h. After incubation, the solution was centrifuged at 10,000 × $g$ for 10 min, and the supernatant was further centrifuged at 200,000 × $g$ at 4 °C for 1 h. The pellet was resuspended in 20 mM Tris-HCl pH 7.4, 150 mM NaCl, 10% sucrose (1 ml per gram tissue) and incubated with 0.5 mg/ml DNase I (TIANGEN, China) and RNase A (Beyotime, China) at 37 °C for 1 h. After incubation, the sample was centrifuged at 6,000 × $g$, the supernatant was further centrifuged at 200,000 × $g$ at 4 °C for 1 h, and the pellet was resuspended in 10 mM Tris-HCl pH 7.4, 150 mM NaCl, 10% sucrose (1 ml per gram tissue). This centrifugation cycle was repeated once, and the final pellet was resuspended in 20 mM Tris-HCl, pH 7.4, 150 mM NaCl (200 μl per gram tissue) and stored in −80 °C for further use. For cryo-EM data collection, the stock solution was further centrifuged at 100,000 × $g$ at 4 °C for 1 h, and the pellet was resuspended in 20 mM Tris-HCl, pH 7.4, 150 mM NaCl (20 μl per gram tissue). The resuspended sample was centrifuged at 6,000 × $g$ for 10 min, and the supernatant was stored at 4 °C for cryo-EM sample preparation.

To analysis GP2 from the small intestine tissue, the frozen tissues from three donors were weighted and washed with a buffer containing 20 mM Tris-HCl, pH 7.4, 150 mM NaCl at a ratio of 4 ml per gram tissue. The washout solution was collected and centrifuged at 10,000 × $g$ for 10 min at 4 °C, and the supernatant was used for further analysis.

### Negative stain transmission electron microscopy (TEM)

An amount of 2.6 μl sample was applied onto freshly glow-discharged 200 mesh carbon-coated Cu grids (Beijing Zhongjingkeyi Technology Co.) and sedimented for 2 min. The excess solution was blotted off by a piece of filter paper. The grids were further stained with 3.3 μl 2% uranyl acetate for 1 min and wash with an additional 3.3 μl of 2% uranyl acetate. The grids were air-dried for 2 min and imaged using a Talos L120C G2 transmission electron microscope (Thermofisher, USA).

PLOS Biology

**Cryo-EM data collection and processing**

Cryo-EM grids were prepared by applying 2.6 µl of sample solution onto glow-discharged grids (Quantifoil 1.2/1.3 200 mesh) and then plunged frozen into liquid ethane using a Vitrobot Mark IV (Thermo Fisher Scientific). Cryo-EM data were collected at Instrument Analysis Center (IAC), Shanghai Jiao Tong University, on a Titan Krios transmission electron microscope (Thermo Fisher Scientific). The microscope was equipped with a K2 Direct Detection Camera (BioQuantum) and operated with 300 kV acceleration voltage and energy filter of 20 eV. Super-resolution movies were collected with a nominal physical pixel size of 1.05 Å/pixel (0.525 Å/pixel in super-resolution movie frames) and a dose per frame of approximately 1.25 e-/Å$^2$. A total of 32 frames were taken for each movie, resulting in a final dose of approximately 40 e-/Å$^2$ per image. Automatic data collection was performed using SerialEM v.3.8.6 software.

Cryo-EM data processing was performed using cryoSPARC v4.0.1 [25], including motion correction and CTF estimation. Automatic particles picking was performed using the filament tracer program, and particles were extracted with a box size of 320 pixels and an inter-box distance of 32 pixels. In the first round of 2D classification, a total of 1,437,210 particles were used, and 1,101,929 particles were selected for 3D reconstruction with 5 *ab initio* initial models. Subsequently, 3D heterogenous refinement was performed using the 5 initial models as references, and particles from each 3D class were subjected to another round of 2D classification. From these 2D classes, 404,189 particles were selected and used for an additional particle selection cycle, which included *ab initio* 3D reconstruction, 3D heterogenous refinement, and 2D classification. Among the resulting particles, 253,399 were selected for local motion correction and non-uniform refinement. A reconstruction with a resolution of 3.62 Å were generated, and local CTF refinement was performed using these particles. Following this, a final 3D heterogenous refinement was performed with three 3D classes and the 3.62 Å reconstruction as a reference. From this 3D heterogenous refinement, 249,718 particles were selected and used for the last round of non-uniform refinement, resulting in a final resolution of 3.49 Å. The map resolution was estimated using the 0.143 Fourier shell correlation (FSC) resolution cutoff, and detailed data collection and processing statistics were listed in Table 1. The map was sharpened using phenix.autosharpen [26] before atomic model building.

We note that the particles in this data processing pipeline were treated as single particles rather than filaments, and no helical symmetry was utilized during the data processing. Consequently, the final 3D-reconstruction only represents the central region of GP2 filaments and corresponds to approximately two ZP modules of GP2. To assess the helical symmetry of GP2 filaments, we performed an additional round of helical refinement using the 3.49 Å reconstruction as a reference. The helical twist and rise were initially set to 180 degrees and 66.1 Å, and then automatically refined to 177.2 degrees and 66.3 Å, respectively. However, helical refinement was unable to generate maps with resolution as high as that achieved with the single particle method (the highest resolution obtained in helical refinement was 4.4 Å). This could be attributed to the influence of the flexible branch and N-glycans of GP2. Additionally, it is possible that the helical parameters were not strictly consistent across all filaments, contributing to the lower resolution in the helical refinement approach. In Fig 1d, to demonstrate the general organization of ZP modules, we manually expanded the 3.49 Å reconstruction by applying the refined helical symmetry using relion_helix_toolbox [27]. It is important to note that this manually expanded reconstruction was solely used for the purpose of presentation, while the 3.49 Å reconstruction without symmetry expansion was used for atomic model building.

During the cryo-EM data processing, we observed the presence of another filament species that appeared thinner than GP2 filaments (Fig 1a, grey arrows). The 2D classes of these filaments exhibited a distinct pitch of approximately 3.4 nm and a filament width of around 2 nm (Fig 1b), which is consistent with the characteristics of B-form DNA [15]. Therefore, we believe that these thinner filaments correspond to B-DNA filaments extracted from the human pancreas. In addition to the filaments, we also observed the presence of globular particles in our cryo-EM micrographs (Fig 1a, black arrows). These particles could potentially represent a globular protein from the human pancreas that either binds to GP2 or DNA filaments or become enriched by our filament extraction protocol. To analyze these particles, we utilized the blob picker in cryoSPARC and performed 2D classification with extracted particles. However, the resulting 2D classes did not

exhibit high-resolution features of these particles, suggesting that they may have been damaged. This damage could be attributed to suboptimal buffer conditions, especially if the protein in question is a membrane protein, as we did not include detergents after the 10% sarkosyl wash during the initial steps. Further investigations are necessary to optimize the extraction protocol and enhance the enrichment of this protein, allowing for its identification through high-resolution cryo-EM structure determination. Nonetheless, we acknowledge that such endeavors are beyond the scope of this study, which primarily focuses on GP2 filaments.

### Atomic model building

The initial model was built de novo with human GP2 sequence in COOT [28]. The model consisted of two ZP modules and three polypeptide chains were first generated, including an intact ZP module (chain A, residues 221–477, with the loops spanning residues 239–251 and 264–274 not built due to poor cryo-EM densities), a ZPC domain from the previous module (ZPC$_{-1}$, chain B, residues 338–477), and a ZPN domain form the subsequent module (ZPN$_{+1}$, chain C, residues 221–337 without 239–251 and 264–274). N-glycans were added to four glycosylation sites (Asn260, Asn291, Asn342, Asn362) based on the presence of extra densities near these residues (Fig 1b and 1f). The initial model was refined using phenix.real_space_refine [26]. During refinement, non-crystallographic symmetry (NCS) were applied, and the ZP domains located at the center of the cryo-EM map were used as references. Specifically, residues 322–341 of chain A, residues 342–477 of chain B, and residues 221–321 of chain C were used as references for the corresponding residues of other chains. This refinement strategy offered two advantages: (i) it utilized the high-resolution central region of the cryo-EM map for refinement and (ii) it maintained an intact ZP module (chain A) during refinement to ensure the geometry of all covalent bonds. The final model, consisting of two ZP modules, was used for structural analysis and presentation. The central region that used as references during refinement was validated using MolProbity [29].

### Western blotting

Samples were mixed with 25% v/v SDS loading buffer, boiled at 100 °C for 10 min, and separated with 4%–20% Bis-Tris precast gels (GenScript, catalog number M00928). The samples were transferred onto 0.22 μm PVDF membranes (Merk, USA), and the membranes were incubated with gentle rocking in 5% non-fat milk (Sangon Biotech, China) in 1× TBST buffer (Tris-buffered saline with 0.1% v/v Tween 20) for 1 h. Next, the membranes were incubated overnight at 4 °C with either anti-GP2 (111–387) polyclonal antibody (Signalway Antibody, USA, catalog number 30715, dilution factor 1:1000) or anti-GP2 (35–179) monoclonal antibody (Novus, USA, catalog number NBP2–53322, dilution factor 1:200) in NCM universal antibody diluent (NCM, catalog number WB500D). Following incubation, the membranes were washed three times in TBST with gentle rocking for 10 min each. Subsequently, the membranes were incubated at room temperature for 1 h with anti-Rabbit IgG HRP (Cell Signaling Technology, USA, catalog number 7074S, dilution factor 1:2000) or anti-Mouse IgG HRP (Cell Signaling Technology, USA, catalog number 96714S, dilution factor 1:3000) in NCM universal antibody diluent. The membranes were then washed for three additional times in TBST. To visualize the protein bands, ECL chemiluminescence substrate (Tanon, China) was applied to the membranes, and the membranes were imaged using a ChemiDoc MP imaging system (Bio-Rad, USA). All western blotting assays were repeated independently at least three times, and similar results were obtained in each repetition.

### Total RNA extraction and GP2 genotyping

The total RNA was extracted from the pancreas tissue of donor 1 using the SteadyPure Universal RNA extraction Kit (Accurate Biology) according to the operating manual. Briefly, the tissue (10 mg) was weighted, flash-frozen, and homogenized, followed by transfer to lysis buffer (buffer RLS). An equal volume of 70% ethanol was added to the mixture, which was then clarified by centrifugation at 12,000 rpm (14,000 × $g$) for 5 min. Subsequently, the supernatant was applied to a Universal RNA mini column, which was further washed once with buffer RWA and twice with buffer RWB. The total RNA

was eluted using 50–100 μl of RNase-free water. For subsequent PCR and sequencing analyses, the RNA was firstly reverse transcribed using the HiScript III 1st Strand cDNA Synthesis Kit (+gDNA wiper) (Vazyme) with oligo(dT) as the primer. The *gp2* gene was then amplified using the cDNA as the template, noting that *gp2* gene was amplified in two fragments to ease the amplification process. Next, the amplification products were purified and ligated into the pCE2 TA/Blunt-Zero vector using the 5 min TA/Blunt-Zero Cloning Kit (Vazyme). The resulting plasmids were transformed into *E. Coli* TOP10 competent cells and screened using the antibiotic marker kanamycin. Positive clones were selected, and the *gp2* gene was sequenced by Sanger sequencing.

## N-glycosylation analysis with liquid chromatography-tandem mass spectrometry

Samples extracted from the pancreas were separated by SDS–PAGE as previously described. The gels were stained using the eStain protein staining device (GenScript). Gel bands around 105 kDa (which aligned with the band observed in western blot assays) were excised for mass spectrometry analysis (S7a Fig). The proteins trapped in the excised gel bands were reduced through incubating with 5 mM TCEP for 30 min at room temperature. Subsequently, the proteins were alkylated by being mixed with IAA (200 mM) to a final concentration of 10 mM and incubated in darkness for 30 min at room temperature. Next, the proteins were separately digested with chymotrypsin, elastase, pepsin, or trypsin, with a molar ratio of protease to protein set at 1:50 for each digestion. All proteolytic digestions were carried out overnight at 37 °C, except for pepsin, which was conducted in HCl buffer (pH 1.0) for 1.5 h at 37 °C. Subsequent to digestion, the protein digests were passed through a C18 column to remove salts. The elution was collected, dried, and prepared for further analysis.

Protein glycosylation was analyzed using a nanoElute HPLC (Bruker) connected to a timsTOF Pro 2 mass spectrometer (Bruker). The proteolytic digested peptides were dissolved in 0.1% formic acid (FA). A 300 ng sample was injected and separated on a C18 column at a flow rate of 300 nl/min, with solvent A and solvent B composed of 0.1% FA and acetonitrile (ACN) containing 0.1% FA, respectively. The liquid chromatography (LC) gradient was programed as follows: 0–45 min, 4%–22% B; 45–50 min, 22%–35% B; 50–55 min, 35%–80% B; 55–60 min, 80% B. For timsTOF Pro2 settings, precursor ions within an m/z range between 100 and 4,000 with charge states 2–5 were chosen for fragmentation. The target intensity per individual PASEF precursor was set at 2,500, and PASEF was configured for 10 MS/MS scans. The data acquisition was conducted in a data-dependent acquisition (DDA) mode.

The raw data was analyzed by PEAKS GlycanFinder with the following settings. The N-linked peptide score and N-linked glycan score were set to 15, with a precursor tolerance error of 20.0 ppm, fragment tolerance of 0.05 Da, glycan fragment error tolerance of 40.0 ppm, C (Carbamidomethylation) as a fixed modification, and M (Oxidation) and NQ (Deamidation) as variable modifications. These settings were consistent across all searches. The proteolytic cleavage site was individually designated for each of the four digestion samples. When searching with only the GP2 sequences (containing isoform 3 and isoform 4), the −10logP value was set to be greater than or equal to 15; when searching with the whole human protein database, the false discovery rate (FDR) was set to 0.05.

The GP2 regions detected in this study are presented below. The full-length GP2 sequence is displayed, with Val179–Arg181, absent in GP2 isoform 3, enclosed in parentheses. The detected regions are labeled with a bold.

MPHLMERMVGSGLLWLALVSCILTQASA**VQRGYGNPIEASSYGLDLDCGAPGTPEAHV**CFDPCQNYTLL**DEP FRSTENSAGSQGCDKNMSGWYR**FVGEGGVRMSETCVQVHRCQTDAPMWL<u>N</u>GTHPALGDGITNHTACAHWSGNC CFWKTEVLVKACPGGYHVYRLEGTPWCNLRYCT(VPR)**DPSTVEDKCEKACRPEEECLALNSTWGCFCRQDLNSSD VHSLQPQLDCGPR**EIKVKVDK**CLLGGLGLGEEVIAYLRDPNCSSILQTEERNWVSVTSPVQASACRNILERNQTHAIYK NTLSLVNDFIIRDTILNINFQCAYPLDMKVSLQAALQPIVSSLNVSVDGNGEFIVRMALFQDQNYTNPYEGDAVEL**SVES VL**YVGAILEQGDTSRF**NL**VLRNCYATPTEDKADLVKYFIIRNSCSNQRDSTIHVEENGQSSESRFSVQMFMFAGHYDLV**FL HCEIHLCDSLNEQCQPSCSRSQVRSEVPAIDLARVLDLGPITRRGAQSPGVMNGTPSTAGFLVAW**PMVLLTVLLAWLF.

## Proteolytic cleavage assays

The concentration of total proteins (including contaminated proteins) in the GP2 filaments sample extracted from donor 1 was estimated to be 1 mg/ml based on OD280 measurements with a standard extinction coefficient. For enzymatic cleavage, the sample was mixed with elastase (Bioss, China, catalog number D13054) at a final concentration of 0.5 mg/ml and incubated for 20 h at 37 °C. Subsequently, the samples were subjected to analysis using SDS–PAGE, western blot, and negative stain electron microscopy. GP2 filaments that were not subjected to elastase cleavage were also included in the analysis for comparison purposes.

The purity of GP2 sample was initially estimated to be around 3% via comparing the intensity of the 110 kDa band and that of all smeared bands in the SDS–PAGE displayed in S7 Fig. The band intensities were assessed using ImageJ software. In addition, considering that full-length GP2 could potentially resistant denaturation by the SDS loading buffer, leading only a portion of GP2 to appeared as the 110 kDa band (S1 Supplementary Notes (Note 2)), the estimated purity of GP2 could be adjusted to 15%. This adjustment was made because the intensity of the eGP2 band was approximately 5-fold higher than that of the GP2 band, as shown in the western blot in Fig 3a (assessed with ImageJ). We acknowledge that the purity estimation is somewhat approximate, and achieving a precise estimation is challenging in this study. It is worth noting that the imprecise purity assessment of GP2 is unlikely to impact the results of the cleavage assays, although we cannot entirely dismiss the potential influence of contaminated proteins.

## FimH$_L$ expression and purification

The cDNA of FimH$_L$ (FimH lectin domain, containing residues 1–180 with A27V mutation) from UPEC strain UTI89 [30] was synthesized at Sangon Biotech (China), and cloned into pET28a expression plasmid. The plasmid was transformed into *E. coli* BL21 (DE3) cells (Yeasen, catalog number 11804ES80) for protein expression. *E. coli* cells were cultured in LB medium at 37 °C overnight, and were further cultured in mannose-free M9 minimal medium to the optical density at 600 nm (OD600) of 0.8–1. After that, 1 mM isopropyl β-D-1-thiogalactopyranoside (IPTG) was added to induce the protein expression, and the cells were incubated in 30 °C for 20 h. His-tagged FimH$_L$ proteins were purified from the periplasmic extract as previously reported with minor modifications [31]. Briefly, cells were harvested by centrifugation at 10,000 × g for 15 min and the pellet was suspended with ice-cold resuspension buffer (20 mM Tris-HCl, pH 7.4, 20% m/v sucrose, 10 mM EDTA, 0.7 mg/ml lysozyme) at a ratio of 2 ml per gram pellet. Suspended pellet was incubated on ice for 40 min, and then mixed with 500 mM MgCl$_2$ at a ratio of 160 μl per gram pellet. The mixed solution was centrifuged at 13,000 × g for 30 min, and the supernatant was dialyzed with the buffer containing 20 mM Tris-HCl pH 7.4, 500 mM NaCl, 20 mM imidazole at 4 °C for 8 h. FimH$_L$ proteins were purified from the dialyzed supernatant via immobilized metal affinity chromatography and size exclusion chromatography. The elute buffer for immobilized metal affinity chromatography was 20 mM Tris-HCl pH 7.4, 500 mM NaCl, 200 mM imidazole, and the running buffer for size exclusion chromatography was 20 mM Tris-HCl pH 7.4, 150 mM NaCl. Purified FimH$_L$ was concentrated to a final concentration of 3 mg/ml with Amicon ultra-15 centrifugal filters (Millipore) and stored at −80 °C for further use.

## FimH$_L$ binding assay

To obtain fluorescein-conjugated FimH$_L$ proteins, purified FimH$_L$ was diluted to 5 μM with buffer containing 10 mM Na$_2$CO$_3$, 90 mM NaHCO$_3$, 127 mM NaCl, and mixed with 20 μM (final concentration) FITC (Adamas, China). The solution was incubated overnight with gentle rotation at 4 °C, and then 50 mM (final concentration) NH$_4$Cl was added to the solution and incubated for 2 h to stop the reaction. Unconjugated FITC was washed 3–5 times with Amicon ultra-15 centrifugal filters (Millipore, 3 kDa cut-off). The wash buffer contained 20 mM Tris-HCl and 150 mM NaCl. FITC-FimH$_L$ that retained by centrifugal filter was used for interaction assays.

To investigate the interaction between FimH$_L$ and GP2 filaments, GP2 filaments extracted from donor 1 with or without elastase cleavage (considered as 1 mg/ml of total proteins, as described above) were mixed with 4:1 (v/v) 0.4 mg/ml

FITC-FimH$_L$ and incubated at 25 °C for 24 h. FITC-FimH$_L$ with the same final concentration was also incubated alone or with elastase as controls. The samples were washed 3 times with Amicon ultra-0.5 centrifugal filters (Millipore, 100 kDa cut-off), and the retained proteins were analyzed by SDS–PAGE. FITC-FimH$_L$ samples (with or without elastase) not washed by centrifugal filters were also analyzed by SDS–PAGE as controls. FITC-FimH$_L$ was visualized with ChemiDoc MP imaging system (Bio-Rad, USA) with 488 nm wavelength excitation and 519 nm wavelength emission. After that, the gels were stained with Coomassie Blue to visualize all bands in the samples.

### Immunogold labelling

To prepare samples for immunogold labelling, GP2 filaments extracted from donor 1 with or without elastase cleavage were mixed 1:40 (v/v) with 0.05 mg/ml FimH$_L$ and incubated at 25 °C for 24 h. GP2 filaments mixed 1:40 (v/v) with buffer were also incubated as a negative control. After incubation, 3.3 µl of each sample was applied onto glow-discharged 200 mesh carbon-coated Cu grids (Beijing Zhongjingkeyi Technology Co.) and incubated for 5 min, then excess sample solution was blotted off using filter paper. The grids were incubated with blocking buffer (phosphate buffered saline, pH 7.4, 0.1% w/v BSA) for 10 min, and excess buffer were blotted off. FimH antibody (Cusabio Technology, catalog number CSB-PA362,349ZA01ENV) diluted 1:300 in blocking buffer was applied to the grids and incubated for 1 h at room temperature; excess solution was blotted off. Grids were washed 5 times with PBS, and further incubated with 6 nm Colloidal Gold AffiniPure Donkey Anti-Rabbit IgG (Jackson ImmunoResearch Laboratories, diluted 1:20 in blocking buffer) for 30 min. Excess solution was blotted off, and grids were washed 5 times with PBS. The grids were then stained with 3.3 µl of 2% uranyl acetate for 1 min, washed with an additional 3.3 µl of 2% uranyl acetate and air-dried for 10 min. Talos L120C G2 (Thermofisher, USA) microscope was used for imaging.

### Light microscopy of *E. coli* and GP2

*E. coli* TOP10 competent cells (Yeasen, catalog number 11801ES80) were used in this study for GP2-induced aggregation assay. To confirm the production of type 1 fimbriae by this *E. coli* strain, we extracted the filaments from the strain and identified type 1 fimbriae through mass spectrometry and cryo-EM structure determination. Mass spectrometry analysis led to the identification of FimA, the primary building blocks of type 1 fimbriae (S6b and S6c Fig), while the cryo-EM structure of the type 1 fimbriae was resolved at a 2.6 Å resolution. A representative cryo-EM micrograph and 2D class are presented in S6d Fig, and detailed structure study of type 1 fimbriae in *E. coli* TOP10 will be described in a separate study. Bacterial cells were grown in M9 media at 37 °C overnight, and cells were harvested by centrifugation at $2,000 \times g$ for 5 min and the pellet was washed 3 times with glucose-free M9 media. *E. coli* cells were incubated in glucose-free M9 media for 2 h and then harvested by centrifugation at $2,000 \times g$ for 5 min. The pellet was resuspended in fresh glucose-free M9 media to an OD600 of 1 and then mixed with 5:1 v/v GP2 extracted from donor 1 with or without elastase cleavage, or with buffer as a control. For D-mannose competition assays, different concentrations of D-mannose were supplemented to GP2/*E. coli* cell mixture. The mixed solution was incubated at 37 °C for 2 h with shaking. Aliquots of 5 µl from each sample were spotted onto a microscope slide, covered with a coverslip, and imaged with a 100× objective on a Leica TCS SP8 STED 3X microscope. These experiments were repeated independently for at least three times with similar results.

### Supporting information

**S1 Fig. Workflow of cryo-EM structure data processing of GP2 filaments.**
(TIF)

**S2 Fig. Cryo-EM structure refinement of GP2 filaments. a**, Local resolution estimate of GP2 cryo-EM map, colored from red (3.2 Å) to blue (4.4 Å). **b**, The final map generated from single-particle cryo-EM reconstruction, along with the central ZP module used as references in structure refinement and validation. **c**, FSC curves between two half-maps (left

 

panel) and the cryo-EM reconstruction and refined atomic model (right panel). **d**, Angular distribution of particles used in the final reconstruction. **e**, Potential positions of branches estimated by superimposing the alpha-fold model of GP2 with the cryo-EM structure determined in this study. The D10C and EGF-like domains, along with the loops composed of residues 239–251 and 264–274 of the alpha-fold model, are shown and colored in cyan and red, respectively. **f**, Cryo-EM map and atomic model in representative regions. **g**, Detailed demonstration of flexible loop region in ZPN domain (pink, upper panel), with the same region overlayed with the cryo-EM map (white), the branch (cyan), and flexible loops (red) from the alpha-fold model. Predicted di-sulfur bonds are indicated by black double lines.
(TIF)

**S3 Fig. Secondary structure arrangements of GP2 filaments. a**, Sequence alignment of human GP2 (top) and UMOD (bottom). Secondary structures of each protein are aligned with their amino acid sequences, shown in black for GP2 and grey for UMOD. β-strands are indicated with arrows, loops with solid lines, and flexible loops with dashed lines. N-glycosylation sites observed on filament structures are marked with red triangles, and the elastase and hepsin cleavage sites of UMOD are indicated with scissor marks. **b**, Cryo-EM structure of GP2 filament, with β-strands labeled. **c**, Structure superimposition of GP2 and UMOD in the flexible loop region. **d,** structural alignment of cryo-EM structure of GP2 filament and the AlphaFold model of full length GP2, the linker from a given ZP module is colored in blue and the ZPN and ZPC from the previous and subsequent ZP modules are colored in pink (ZPN$_{+1}$) and green (ZPC$_{-1}$), respectively. The AlphaFold model is colored in cyan.
(TIF)

**S4 Fig. Inter-subdomain interactions of GP2 filaments. a–b**, Interactions between the ZP-linker (blue) and the preceding ZPC domain (ZPC$_{-1}$, green) and the subsequent ZPN domain (ZPN$_{+1}$, pink) in general views **(a)** and detailed views **(b)**. In panel **a**, ZPN$_{+1}$ and ZPC$_{-1}$ are represented as surfaces, colored based on the molecular lipophilicity potential, ranging from most hydrophilic (dark cyan) to most hydrophobic (dark goldenrod). **c**, Detailed interactions between ZPN (blue) and ZPC$_{-1}$ (green, left panel) and between ZPC$_{-1}$ (green) and ZPN$_{+1}$ (pink, right panel). The electrostatic interactions shown in panels **b and c** involve residues Asp325, His464 and Arg399 (panel **b**, top left), Arg290, Asp467, and Glu471 (panel **c**, left), Arg310 and Asp427 (panel **c**, right), and Asp306 and Arg426 (panel **c**, right).
(TIF)

**S5 Fig. Analysis of proteolytic cleavage of GP2 filaments. a–d**, Residue range predictions of each band in **(a)** pancreas-derived GP2 without elastase cleavage (first lane in Fig 3a), **(b)** pancreas-derived GP2 with elastase cleavage (second lane in Fig 3a), **(c)** small-intestine-derived GP2 from donor 2 (fourth lane in Fig 3a), or **(d)** donor 3 (fifth lane in Fig 3a). The contrast of the western blotting has been manually adjusted for better presentation of the bands, and the unadjusted blots are shown in Fig 3a and in S1 Raw Images. Since the band patterns in donor 4 are similar to those of donor 3, we used donor 3 to represent both donors. For detailed analysis, please refer to S1 Supplementary Notes (Notes 1–3) and S2 Table. **e**, The upper panel displays the amino acid sequence of GP2, 181–480, with color coding as described in S1 Supplementary Notes (Note 2). The lower panel shows a schematic of GP2, with N-glycosylation sites labeled and colored in grey, and potential elastase cleavage sites labeled and colored in red. **f**, The structure of GP2 filament with potential elastase cleavage sites labeled and colored in red. The cryo-EM model is colored in white, and the missing parts are complemented by the alpha-fold model, colored in grey.
(TIF)

**S6 Fig. Additional experiments on GP2 induced bacterial aggregation. a**, GP2-induced bacterial aggregation can be inhibited by the competition of D-mannose. Representative light microscopy images of GP2-mediated *E. coli* cell aggregation, incubated without or with the indicated concentration of D-mannose. The scale bar represents 10 μm. **b**, The amino acid sequence of *E. coli* FimA, with the signal peptide colored in grey and unique peptides detected by LC–MS/MS

colored in red. **c**, The MS/MS spectrum of two of the detected unique peptides from *E. coli* FimA. CA, carbamidomethylation. **d**, A representative cryo-EM micrograph and 2D class of type 1 fimbriae extracted from *E. coli* TOP10. The scale bar represents 50 nm.
(TIF)

**S7 Fig. Mass spectrometric analysis of N-glycans on Asn122.** a, SDS–PAGE gel of GP2 filaments extracted from donor 1. The dashed box indicates the gel region corresponding to the approximately 105 kDa band from the GP2 western blot (Fig 3a), which was excised for LC–MS/MS analyses. The original gel can be found in S1 Raw Images. **b**, (Upper panel) LC–MS/MS spectra of the GP2 glycopeptide containing Asn122, with its sequence shown in the top left corners. Detected peptide-backbone fragment ions are labeled as red and blue lines. (Lower panel) LC–MS/MS spectra of the glycopeptide detected using GP2-free database (from a glycoprotein other than GP2), which exhibited a similar mass to the GP2 glycopeptide shown in the upper panel. CA, carbamidomethylation. N-glycan structures are depicted following the Consortium for Functional Glycomics (CFG) notation: blue square, N-acetylglucosamine; green circle, mannose; yellow circle, galactose; red triangle, fucose. (The legend also applies to S8–S15 Figs).
(TIF)

**S8 Fig. Analysis of glycosylation sites on Asn134.**
(TIF)

**S9 Fig. Analysis of glycosylation sites on Asn204.**
(TIF)

**S10 Fig. Analysis of glycosylation sites on Asn216.** DE, deamidation.
(TIF)

**S11 Fig. Analysis of glycosylation sites on Asn260.**
(TIF)

**S12 Fig. Analysis of glycosylation sites on Asn291.**
(TIF)

**S13 Fig. Analysis of glycosylation sites on Asn342.** OX, oxidation.
(TIF)

**S14 Fig. Analysis of glycosylation sites on Asn362.** The lower panel displays LC–MS/MS spectra of the glycopeptide detected using GP2-free database (from a glycoprotein other than GP2), which exhibited a similar mass to the GP2 glycopeptide shown in the upper panel.
(TIF)

**S15 Fig. Analysis of glycosylation sites on Asn420.**
(TIF)

**S1 Table. Primers used for GP2 genotyping and the genotyping results.**
(PDF)

**S2 Table. Bands observed in western blotting shown in Figs 3a and S5a–S5d, and the plausible predictions of the residue range for each band.**
(PDF)

**S3 Table. N-Glycan types detected in this study when using GP2 sequence.**
(PDF)

                                                         

**S1 Raw Images.** Uncropped version of blots and gels shown in Figs 3, S5a–S5d and S7a.
(PDF)

**S1 Supplementary Notes.** Supplementary Notes 1–4.
(DOCX)

## Acknowledgments

The authors thank for cryo-EM data collection at the Instrument Analysis Center (IAC), Shanghai Jiao Tong University. The authors acknowledge the National Facility for Translational Medicine (Shanghai) for support. The authors thank XJ Wang for the help in plasmid construction.

## Author contributions

**Conceptualization:** Jianting Han, Qin Cao.

**Formal analysis:** Jianting Han, Qin Cao.

**Funding acquisition:** Fei Zhang, Qin Cao.

**Investigation:** Jianting Han, Meinai Song, Yijia Cheng.

**Methodology:** Jianting Han, Meinai Song.

**Resources:** Wei Gong, Fei Zhang.

**Supervision:** Fei Zhang, Qin Cao.

**Validation:** Fei Zhang, Jianting Han.

**Writing – original draft:** Qin Cao.

**Writing – review & editing:** Jianting Han, Meinai Song, Yijia Cheng, Wei Gong, Fei Zhang, Qin Cao.

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
