## [Editor Report · Decision Letter 0]

Dear Dr Han,

Thank you for submitting your manuscript entitled "Molecular insights into human glycoprotein 2 (GP2) filament formation and its antibacterial function in the proteolytic environment" for consideration as a Research Article by PLOS Biology. Please accept my sincere apologies for the delay in getting back to you as we consulted with an academic editor about your submission.

Your manuscript has now been evaluated by the PLOS Biology editorial staff, as well as by an academic editor with relevant expertise, and I am writing to let you know that we would like to send your submission out for external peer review.

Once your full submission is complete, your paper will undergo a series of checks in preparation for peer review. After your manuscript has passed the checks it will be sent out for review. To provide the metadata for your submission, please Login to Editorial Manager (https://www.editorialmanager.com/pbiology) within two working days, i.e. by Apr 11 2025 11:59PM.

Kind regards,

Richard

Richard Hodge, PhD

rhodge@plos.org

PLOS

---

## [Editor Report · Decision Letter 1]

Dear Dr Cao,

Thank you for your continued patience while we considered your manuscript "Molecular insights into human glycoprotein 2 (GP2) filament formation and its antibacterial function in the proteolytic environment" for publication as a Research Article at PLOS Biology. Please accept my sincere apologies for the delay in getting back to you as we obtained the previous reviews and reviewer identities from Nature Communications. This revised version of your manuscript has been evaluated by the PLOS Biology editors and the Academic Editor.

Based on our Academic Editor's assessment of your revision and responses to the previous rounds of review at Nature Communications, I am pleased to say that we are likely to accept this manuscript for publication. I have provided some specific comments from the Academic Editor below my signature (labelled ‘Comments from the Academic Editor’) where we think it would be useful to include an estimate of the purity of the GP2 protein preparations in a revised version.

In addition, after discussions with the rest of the editorial team, we would like to consider your manuscript as a Short Report at the journal (https://journals.plos.org/plosbiology/s/what-we-publish#loc-short-reports), given the scope of the study and the previous comments from Reviewer #1. I will change the article type on your behalf upon resubmission.

*IMPORTANT*

Finally, I would be grateful if you could please make sure to address the following editorial and data-related requests that I have provided below (A-H):

(A) We routinely suggest changes to titles to ensure maximum accessibility for a broad, non-specialist readership. In this case, we would suggest a minor edit to the title, as follows. Please ensure you change both the manuscript file and the online submission system, as they need to match for final acceptance:

“Structure of human glycoprotein 2 reveals mechanisms underlying filament formation and adaption to proteolytic environments in the digestive tract”

(B) Please include information about the form of consent (written/oral) given for research involving human participants in the ethics statement provided in the Methods section

(C) Please include the specific approval number issued by the Institutional Review Board of Xinhua hospital in the ethics statement provided in the Methods section.

(D) Thank you for providing the structural data in the PDB and EMDB databases (8XC5 and EMD-38237). However, we note that the data is currently on hold for release. We ask that you please make the structures publicly available at this stage before publication.

(E) In addition, please ensure that the LC-MS/MS raw data deposited in PRIDE is made publicly available and provide the accession number/URL of the deposition in the Data Availability Statement in the online submission form.

(F) Please also ensure that each of the relevant figure legends in your manuscript include information on *WHERE THE UNDERLYING DATA CAN BE FOUND*, and ensure your supplemental data file/s has a legend.

(G) We require the original, uncropped and minimally adjusted images supporting all blot and gel results reported in the following Figures:

Figure 3A, 3C, S6A-D, S8

We will require these files before a manuscript can be accepted so please prepare and upload them now. Please carefully read our guidelines for how to prepare and upload this data: https://journals.plos.org/plosbiology/s/figures#loc-blot-and-gel-reporting-requirements.

We noted that the uncropped gels for Figure 3A are already provided in the Supplementary file as Figure S5. We ask that this is removed and included in a specific supplementary information file named S1_raw_images.

(H) Per journal policy, if you have generated any custom code during the course of this investigation, please make it available without restrictions. Please ensure that the code is sufficiently well documented and reusable, and that your Data Statement in the Editorial Manager submission system accurately describes where your code can be found.

We expect to receive your revised manuscript within two weeks.

*Published Peer Review History*

*Press*

Best regards,

Richard

Richard Hodge, PhD

rhodge@plos.org

COMMENTS FROM THE ACADEMIC EDITOR

Regarding Comment 3 of Reviewer 1, it would be helpful for the authors to state in the manuscript what percentage of the total protein is GP2. Given that they quantitate total protein to determine the protein:elastase ratio in the experiment, this number should be available if they have some ability to estimate GP2 concentration. Adding this percentage would also give the authors and opportunity to acknowledge that while unlikely, the effects of contaminating proteins on their assays cannot be completely ruled out.

---

## [Editor Report · Decision Letter 2]

Dear Qin,

On behalf of my colleagues and the Academic Editor, Ann Stock, I am pleased to say that we can accept your manuscript for publication, provided you address any remaining formatting and reporting issues. These will be detailed in an email you should receive within 2-3 business days from our colleagues in the journal operations team; no action is required from you until then. Please note that we will not be able to formally accept your manuscript and schedule it for publication until you have completed any requested changes.

PRESS

Best wishes, 

Richard

Richard Hodge, PhD

rhodge@plos.org

PLOS
